# Oxidation of Thin Titanium Films: Determination of the Chemical Composition of the Oxide and the Oxygen Diffusion Factor

**Sergey Y. Sarvadii \*, Andrey K. Gatin, Vasiliy A. Kharitonov, Nadezhda V. Dokhlikova, Sergey A. Ozerin, Maxim V. Grishin and Boris R. Shub**

N.N. Semenov Federal Research Center for Chemical Physics, Russian Academy of Sciences (FRCCP RAS) Kosygina str. 4, 119991 Moscow, Russia; akgatin@yandex.ru (A.K.G.); vch.ost@mail.ru (V.A.K.); dohlikovanv@gmail.com (N.V.D.); sergeoz@yandex.ru (S.A.O.); mvgrishin68@yandex.ru (M.V.G.); bshub@mail.ru (B.R.S.)

**\*** Correspondence: sarvadiy15@mail.ru

**Abstract:** The morphologies and local electronic structures of titanium coatings deposited on the surfaces of highly oriented pyrolytic graphite were determined. Chemical compositions of the oxides formed on the coating surfaces were established. A theoretical model was developed describing the changes in the $TiO_x$ oxides $(1.75 < x < 2)$ band gap depending on the duration and temperature of the titanium film annealing procedure in oxygen. The effective activation energy of oxygen diffusion in $TiO_x$ $(1.75 < x < 2)$ was determined, and the pre-exponential factor of the diffusion coefficient was estimated.

**Keywords:** titanium oxide; Magneli phase; band gap; diffusion; scanning tunneling microscopy; scanning tunneling spectroscopy

---

## 1. Introduction

The physicochemical properties of a catalytic system consisting of nanoparticles deposited on a conductive substrate can be controlled by changing the structure of the oxide layer that covers the surface of the substrate [1,2]. With this approach, reliable information on the chemical composition, morphology, and electronic structure of the surface oxide film is of particular importance. The thickness of the oxide layer, its dielectric constant and the electron work function for the substrate material determine the efficiency of charge transfer between the substrate and nanoparticles and affect the mechanisms of chemical reactions accordingly. Both the relatively small electron work function for pure titanium (3.95 eV) [3] and the narrow band gap (<0.1 eV) for a number of its oxides [3–5] can provide conditions for efficient interaction of nanoparticles with the substrate metal even in the case of a sufficiently thick oxide layer separating them. This makes titanium a promising material from the standpoint of creating model catalytic systems.

A large number of works have been devoted to studying the initial oxidation stages of titanium single crystals [6–9], polycrystalline ribbons [10–13] and thin titanium films [14–19]. Nevertheless, significant differences still remain in the interpretation of data on the chemical composition of the oxide growing during the initial oxidation stage and its distribution in the volume of the substrate. In particular, in a number of works, it is assumed that at the initial stages of adsorption, oxygen penetrates into the subsurface layer and only after that is chemisorbed on the surface [6,11,14,20]. On the other hand, there are works [8,10,21,22] in which the authors believe that oxygen initially accumulates on the surface. There are also disagreements regarding the stoichiometric composition of the oxides formed during the oxidation process. In some works, it is pointed out that $TiO_2$ grows

on the surface at large exposure to oxygen [7,23], while in other works [11,15,24], the formation of TiO and $Ti_2O_3$ oxides is assumed; additionally, there are other authors that declare the formation of a mixture of various oxides [6,8,12,13,25]. In [12,19], it is suggested that the composition of the oxides on the surface is affected by surface pretreatment, the oxygen pressure at which oxidation occurs [26], and the substrate temperature [13]. In contrast, [27] indicates that the formation of titanium oxides is practically independent of oxygen pressure. In [28], data are presented that show both temperature and oxygen pressure are the determining factors in the formation of various oxides on the surface of an ultrathin titanium film deposited on Pt (111). This work is an excellent example of multifaceted approach to the investigation of the oxidation of ultrathin metal films and the data presented in it are quite exhaustive in describing the chemistry of the $TiO_x$ phases in the case of the $TiO_x$|Pt(111) system.

There is little data available on the electronic structure of various titanium oxides. It is known that $Ti_2O$ and TiO oxides are conductors, and $TiO_2$ is a semiconductor with a band gap of 3.2–3.5 eV, depending on the crystal structure (rutile and anatase) [3,5,29,30]. Furthermore, the band gap of $Ti_2O_3$ oxide is very narrow (<0.1 eV) [4]. Information on the electronic structure of $Ti_nO_{2n-1}$ oxide phases where n = 4–9 (Magneli phases) is either absent or results from quantum-chemical simulations, so great care must be taken when correlating them with experimental data [31]. It is also necessary to take into account that the electronic structure of thin oxide films is sensitive to various spatial factors such as film thickness and grain size [32].

This work is devoted to the determination of the chemical composition of the oxide layer formed on the surface of a titanium film as a result of its interaction with oxygen at various temperatures. The morphologies and electronic structures of the oxidized titanium coatings are established by scanning tunneling microscopy and spectroscopy, and oxygen diffusion factors are estimated.

## 2. Experimental

Titanium coatings were synthesized by a thermo-resistive vacuum deposition technique. Briefly, a piece of titanium wire was placed in a tantalum evaporator, after which the vacuum chamber was evacuated to a residual gas pressure of $P = 10^{-6}$ Torr. Then, the evaporator was heated to a temperature close to the melting temperature of titanium by passing alternating current through it. A plate (7 mm × 7 mm × 1 mm) of highly oriented pyrolytic graphite (HOPG) was placed in the stream of evaporating titanium for 15–20 seconds. After titanium deposition, the sample was exposed to oxygen. If one needed oxidation to be carried out at room temperature, i.e., without annealing of the sample, the sample was removed from the setup in air ($P_{O2}$ ~ 150 Torr).

To anneal the sample in oxygen, $O_2$ was injected into the setup to a pressure of $P_{O2}$ ~ 100–200 Torr, and the sample was heated to 770 K and exposed to $O_2$ for a given time. The sample was heated by a resistive heater, and the temperature was determined using a chromel–alumel thermocouple.

Data on the elemental composition of the prepared samples were obtained by Auger spectroscopy.

The structure of the titanium coating was studied using scanning tunneling microscopy and spectroscopy (STM/STS) methods, which made it possible to determine the morphologies and local electronic structures of the obtained titanium films. The morphological structures of the samples were investigated by STM operating in a constant current topographical mode, and the electronic structures were studied by measuring the dependency of the tunneling current on the voltage applied to the STM probe. Investigations were performed with a setup equipped with an ultrahigh-vacuum scanning tunneling microscope (UHV VT STM, Omicron NanoTechnology, Taunusstein, Germany), a CMA-100 Auger spectrometer (Omicron NanoTechnology, Taunusstein, Germany) and pipelines for gas input. The residual gas pressure in the STM chamber did not exceed $2 \times 10^{-10}$ Torr. The tungsten STM probes used for the experiments were prepared by electrochemical etching in an aqueous solution of 0.1 M potassium hydroxide and treated by argon–ion sputtering under ultrahigh vacuum conditions to remove the oxide layer.

## 3. Results and Discussion

The elemental composition of the surface of freshly prepared samples was determined by Auger spectroscopy and showed the presence of carbon, titanium and oxygen. For comparison, spectra were obtained for a titanium wire and for a stoichiometric $TiO_2$ film synthesized on the surface of a bulk titanium plate according to a procedure described in [33]. As shown in Figure 1, the peak positions of titanium coincide in all three cases.

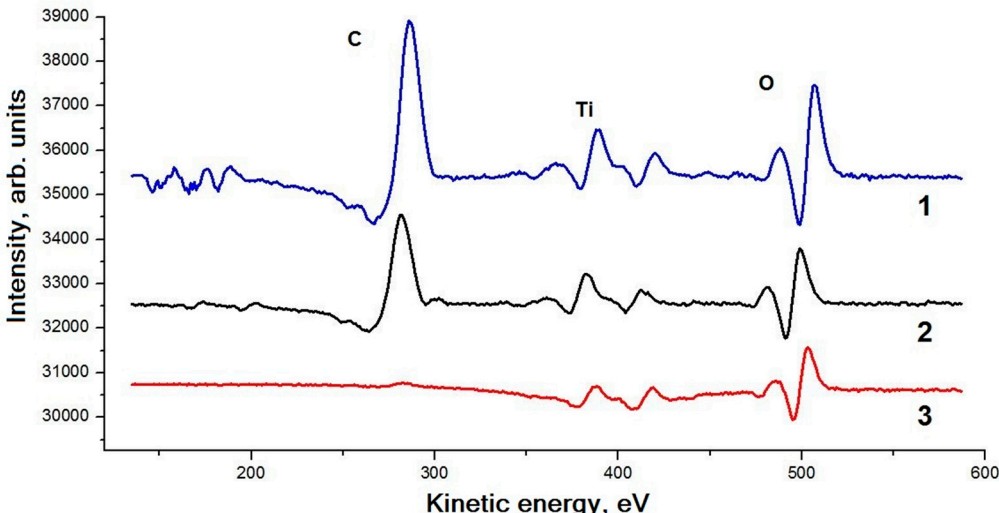

**Figure 1.** Auger spectroscopy results: 1, a titanium dioxide film synthesized according to a method in [31]; 2, a titanium wire; and 3, a freshly prepared titanium coating sample.

After the interaction with oxygen, the titanium coating samples were placed in an STM vacuum chamber, where the morphology and electronic structure of each sample surface were determined using topographic and spectroscopic measurements. It is known [34] that a nanocontact formed by a metal sample and a conducting STM tip corresponds to an S-shaped dependence of the STM tunneling current (current-voltage dependence, CVD). Oxidation of the sample surface can lead to a significant decrease in the density of states of the metal in the vicinity of the Fermi level, which leads to an appearance of a zero-current region within the S-shaped curve. The extent of this region corresponds to the band gap of the material under the STM tip up to a dimensional factor [35–38]. Thus, the shape of the CVD curve is an indicator signaling variations in the local chemical composition of the surface.

As a result of STM measurements, it is found that the coatings consist of titanium grains in close contact with each other. The titanium grains have a characteristic lateral size of approximately 30 nm and a height of 3–6 nm (see Figure 2). Additionally, the presence of both smaller (up to 5 nm) and larger titanium nanoparticles is revealed.

Measurements of the extent of the zero-current region of the CVD curves at various points of the sample surface show that most of the surface demonstrates a band gap with energy depending on the oxidation conditions. These measurements of local band gap energies $E_g^{loc}$ make it possible to map the oxide phase distribution over the surface of the titanium coating (see Figure 3). The oxide phase distribution in the case of oxidation in air is more uniform than in the case of annealing.

For both sections shown in Figure 3, histograms of the occurrence frequency for various $E_g^{loc}$ values were plotted (see frequency distribution histograms in Figure 4). Comparing the obtained histograms, one can note that both distributions are unimodal. In the case of oxidation in air, the distribution maximum is rather narrow, and upon annealing, it noticeably broadens and shifts toward high band gap energies. The position of the distribution maximum corresponds to the most probable band gap energy $E_g$ under the given oxidation conditions. The results of the measurement of $E_g$ in the case of various oxidation conditions are presented in Table 1.

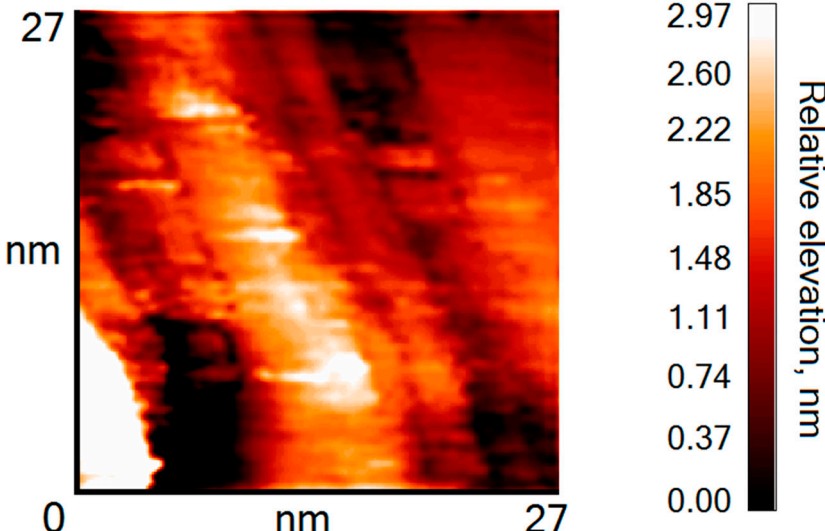

**Figure 2.** Surface topography of the titanium coating sample after oxidation in air ($t$ = 720 h, $T$ = 300 K).

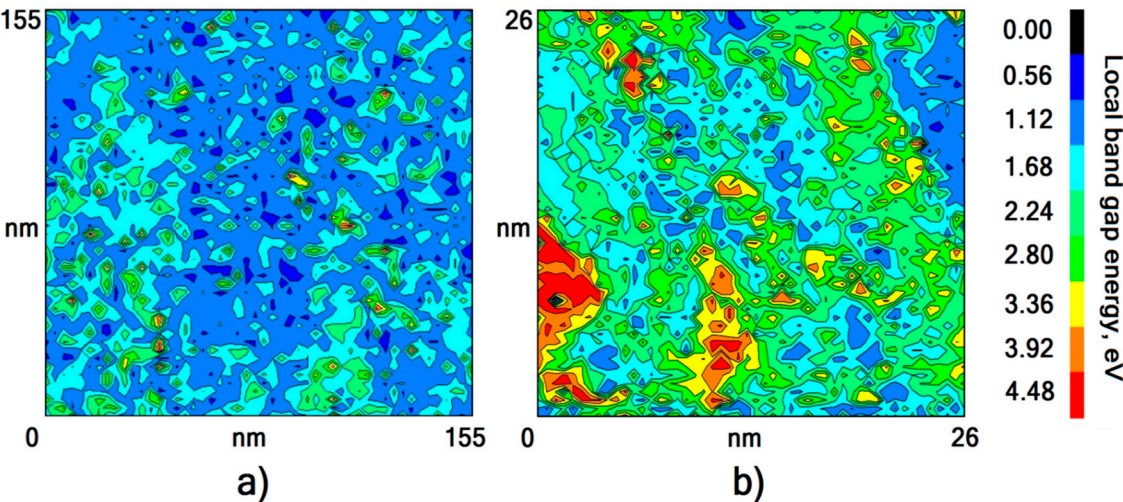

**Figure 3.** Map of the oxide phase distribution over the titanium coating surface. Different colors indicate areas with different local band gap values $E_g^{loc}$: (**a**), oxidation in air ($t$ = 720 h, $T$ = 300 K); (**b**) oxidation during annealing ($t$ = 13 h, $T$ = 770 K).

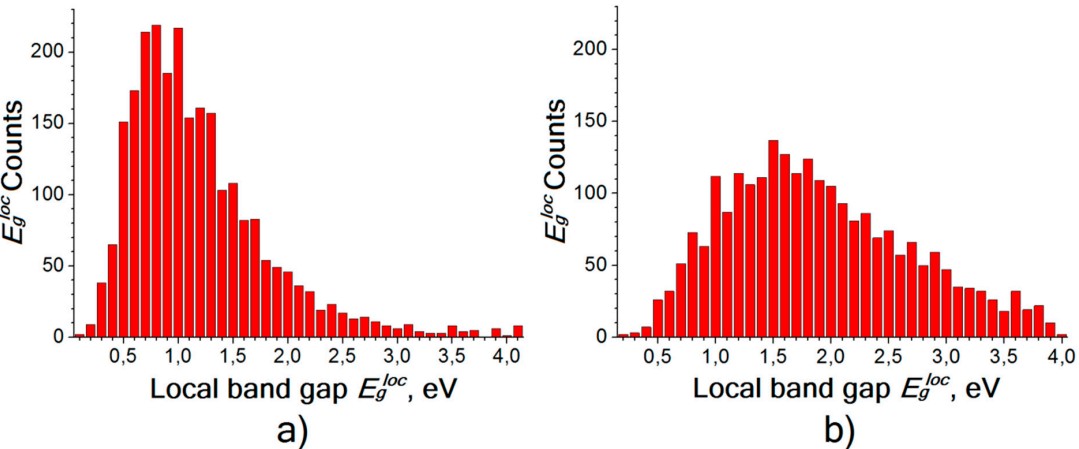

**Figure 4.** Histograms of the frequency distribution of $E_g^{loc}$ values: (**a**), oxidation in air ($t$ = 720 h, $T$ = 300 K); (**b**), oxidation during annealing ($t$ = 13 h, $T$ = 770 K).

**Table 1.** Results of $E_g$ measurements for various oxidation conditions. $T$ is the annealing temperature, $t$ is the annealing duration of the sample in oxygen, $E_g$ is the most probable band gap energy under the selected oxidation conditions.

| Sample, Nos | $T$, K | $t$, h | $E_g$, eV |
|:---:|:---:|:---:|:---:|
| 1 | 300 | 720 | 0.8 |
| 2 | 770 | 4 | 0.8 |
| 3 | 770 | 13 | 1.8 |

## 4. Determination of the Oxide Chemical Composition

For titanium oxides, the following band gap energies are known depending on the oxygen content in the oxide (Table 2).

**Table 2.** Oxygen content (per titanium atom) and band gap energy $E_g$ for various titanium oxides.

| Oxide | O:Ti | $E_g$, eV | Reference |
|:---:|:---:|:---:|:---:|
| $Ti_2O$ | 0.5 : 1.0 | 0.0 | [29] |
| TiO | 1.0 : 1.0 | 0.0 | [3] |
| $Ti_2O_3$ | 1.5 : 1.0 | <0.1 | [4] |
| $TiO_2$ | 2.0 : 1.0 | 3.0–3.2 | [5,30] |

Obviously, it is not possible to detect the presence of $Ti_2O$, TiO, and $Ti_2O_3$ oxides on the surface of the sample since they are conductors or have a band gap that is too narrow. Moreover, it can be argued that the formation of $TiO_2$ does not occur on the surface of titanium grains, with the exception of some individual points. Thus, the measured $E_g$ values (Table 1) on the surface of oxidized titanium coatings do not correspond to any stoichiometric titanium oxide, and the coincidence of the $E_g$ values for samples 1 and 2 is random. It is noteworthy that the band gap of TiC is 0.4 eV [39], which also does not correspond to the measured $E_g$ values. The results of quantum chemical calculations in [31] show that the band gap of 1.8 eV can correspond to $Ti_3O_5$ in the case of an isolated 2D structure, but this interpretation is not applicable under the conditions of our experiment.

The observed discrepancy between the literature data and the measurement results has two possible explanations. The first is that an oxide is formed on the surface of the titanium coating that is different in its stoichiometry from the ones shown in the Table 2. Indeed, according to [40], in the range between $Ti_2O_3$ and $TiO_2$, there is a group of homologous phases $Ti_nO_{2n-1}$, where n = 4–9 (Magneli phases). For example, in articles [41,42], it is shown that a band gap of 0.25–0.29 eV appears at low temperatures for $Ti_4O_7$, which is conductive at room temperature. However, another explanation is possible, namely, a thin layer of stoichiometric $TiO_2$ forms on the surface of metallic titanium. The $TiO_2$ electronic structure is distorted due to interaction with metallic titanium, and as a result, the observed band gap may be less than real one.

To determine the thickness of stoichiometric $TiO_2$ required for observation of the band gap of 0.8 eV, a calculation was made. According to [43], the dependence of the current $I_{ox}$ of electrons tunneling through an oxide layer of thickness $d$ and a vacuum gap of width $z$ on a bias voltage $U$ (see Figure 5) can be written as:

$$I_{ox}(z,d,\ U) = B \int_{E_F-U}^{E_F} d\varepsilon |\varepsilon - E_F + U| [Exp\{-\kappa_e(\varepsilon,\ z,\ d,\ U)d\} + Exp\{-\kappa_h(\varepsilon,\ z,\ d,\ U)d\}]^2$$
$$\times Exp\left\{-2\int_0^z dx \sqrt{2\left(-\varepsilon - \frac{U-U_{ox}(U,\ d,\ z)}{z}x - \frac{1}{4(z-x+b)}\right)}\right\}$$

(1)

where $E_F$ is the Fermi level of the electrode (tungsten tip), $U_{ox}$ is the voltage drop in the oxide layer of thickness $d$ and dielectric constant $\chi$, and $\kappa_e$ and $\kappa_h$ are the tunneling quasimomenta (crystal momenta) of the electron and hole, respectively. The constant $b$ was obtained from the results of theoretically processing the tunneling current in vacuum, $b \approx 5$ [43].

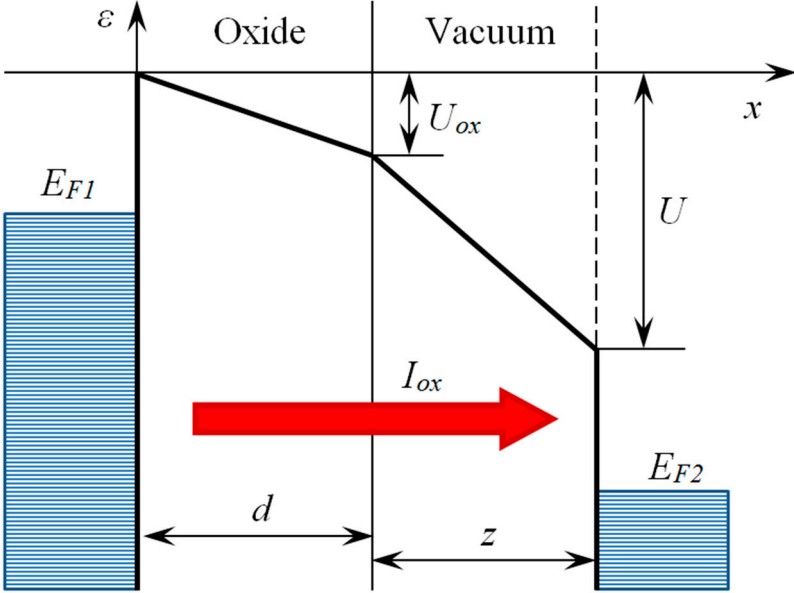

**Figure 5.** An energy diagram explaining the process of electron tunneling through an oxide layer of thickness $d$ and a vacuum gap of width $z$ depending on a bias voltage $U$.

The calculation procedure consists of minimizing the quadratic deviation of the theoretical curve from the experimental curve in terms of $z$ and $d$. Since the factor $B$ weakly (not exponentially) depends on the desired parameters and cannot be accurately calculated, relative currents $I_{ox}(z, d, U)/I_{ox}(z, d, U_{max})$, where $U_{max} = 1.6$ V are taken for minimization. The calculation is carried out under the assumption that the oxide that forms on the titanium surface is $TiO_2$ with a band gap of $E_g = 3.2$ eV and $\chi = 100$ [44,45]. For calculations, a Wolfram Mathematica 11 software package is used.

The minimization results show that for sample No. 2, the maximum agreement between the calculated and experimental curves is achieved at $z = 3.9$ Å, $d = 4.5$ Å. It is known that the unit cell parameters for $TiO_2$-rutile are $a = 4.60$ Å, $c = 2.96$ Å [46], and for $TiO_2$-anatase $a = 3.78$ Å, $c = 9.50$ Å [47], which is relatively similar to the obtained oxide layer thickness $d = 4.5$ Å. That is, the calculated $TiO_2$ thickness is such that for small oxide thickness values, we cannot talk about any specific crystalline structure—it simply will not have a short-range order—and for large oxide thickness values, the observed band gap will increase. It is known from the literature that oxygen actively dissolves in titanium and easily penetrates through the formed oxide layer [27]. Thus, the existence of such a thin oxide layer with perfect structure and a clear metal-oxide interface is highly doubtful—we cannot consider our material to be a mixture of $TiO_2$ fractions and Ti metal fractions. The obtained data and calculation results allow us to conclude that the surface of titanium film after 4 h of annealing in $O_2$ is indeed coated with a layer of a $TiO_x$ oxide with a band gap of 0.6–0.8 eV. Since $Ti_4O_7$ oxide is a conductor at room temperature [41], we can say that the formed oxide most likely corresponds to $TiO_x$, where $1.75 < x < 2$. Note that with an increase in the annealing duration, band gap broadening occurs, and the oxygen content in the oxide approaches that in dioxide. Indeed, after 13 h of annealing in oxygen (sample No. 3), it was possible to observe, at many points on the surface, a band gap energy of 3.0–3.2 eV that corresponds to the literature data for $TiO_2$ [5,30]; but at the same time, there are points at which the measured local $E_g$ exceeds these values. The following aspects should be noted. First, the literature data for titanium dioxide $E_g$ may vary depending on the measurement method. For example,

the article [48] gives higher values of 3.7-3.8 eV. Second, large values of the local $E_g$ measured by us can be a result of such phenomena as the Coulomb blockade or charging. The presence of a clear peak in the histogram in the region of 3–4 eV would clearly distinguish these two aspects, but, unfortunately, this peak is not observed under the conditions of our experiment. That is, we cannot estimate the amount of dioxide formed, but part of the values we have measured indicates that titanium dioxide is formed on the surface. This result above shows that the increase in the band gap observed by us is not jump-like, as it should be in the case of successive conversion of different oxide phases, but gradual as in the case of a range of very similar phases. That is, we can consider the band gap of the resulting oxide film to be a function of the amount of oxygen that dissolves in the titanium film and reacts with it. This results in two consequences. First, it is possible to relate the change in the band gap directly with the process of oxygen diffusion in titanium and to establish the parameters of this process. Second, by changing the temperature and duration of the annealing procedure for a sample in oxygen, one can obtain titanium oxide coatings with a desired band gap value.

## 5. Determination of the Oxygen Diffusion Factor in Titanium

One can consider the sorption process in a symmetric plate of thickness $H$, with two symmetrically directed flows from an infinite source. A solution is known for such a problem, that is, the distribution of the sorbate concentration in the plate [49]:

$$C(x,\ t) = C_0\left[1 - \frac{4}{\pi}\sum_{k=0}^{\infty}\frac{1}{2k+1}\times exp\left[-\frac{(2k+1)^2\pi^2Dt}{H^2}\right]\times sin\left[\frac{(2k+1)\pi}{H}x\right]\right]$$

(2)

where $D$ is the coefficient of sorbate diffusion into the plate, $x$ is the coordinate along the plate thickness (see Figure 6), and $C_0$ is the maximum possible concentration of sorbate in the plate as $t \to \infty$. Since the problem is symmetric with respect to the plane $x = H/2$, we can consider the problem only on the segment $x \in (0, H/2)$, which corresponds to our system with a titanium coating; our system is open to oxygen diffusion from one side ($x = 0$) and closed from the other side by a reflective screen ($x = H/2$).

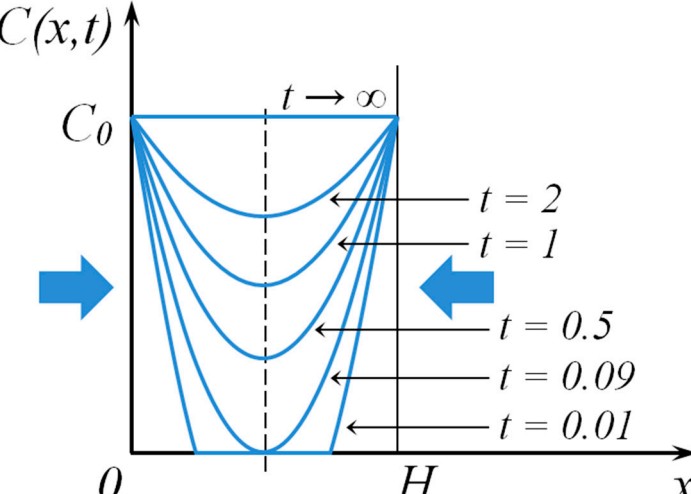

**Figure 6.** Distributions of sorbate concentration $C$ at various times $t$ in a plate of thickness $H$ in the case of symmetrically directed sorbate flows.

The electronic structure of the titanium surface changes as oxygen dissolves in the film and reacts with titanium. It is necessary to find the amount of oxygen $q$ that has penetrated into the titanium film. It should be noted that STM is sensitive to the electronic structure of only a thin layer at the surface of

the coating, i.e., to a titanium layer with thickness $x_0 \ll H/2$. Then, we find the amount of oxygen in the surface layer ($0 < x < x_0 \ll H/2$).

$$q(x_0,\ t) = \int_0^{x_0} C(x,\ t)dx =$$

$$= C_0 x_0 - C_0 H \frac{4}{\pi^2} \sum_{k=0}^{\infty} \frac{1}{(2k+1)^2} \times exp\left[-\frac{(2k+1)^2 \pi^2 Dt}{H^2}\right] \times \left[1 - cos\left(\frac{2k+1}{H}\pi x_0\right)\right] \tag{3}$$

For a sufficiently large time $t$, we can restrict the series with k = 0 [46].

If the requirement $\pi x_0/H \ll 1$ is met, then the following is obtained: $1 - cos\left(\frac{2k+1}{H}\pi x_0\right) = 1 - cos\left(\frac{\pi x_0}{H}\right) \approx \frac{1}{2}\frac{\pi^2 x_0^2}{H^2}$.

Subsequently, one can write the following:

$$q(x_0, t) = C_0 x_0 \left[1 - 2\frac{x_0}{H} \times exp\left(-\frac{\pi^2 Dt}{H^2}\right)\right] \tag{4}$$

In the above, note that $q(x_0,\ t) \to q_0 = C_0 x_0$, when $t \to \infty$.

The measured band gap depends on the amount of oxygen in the titanium layer with a thickness of $x_0$. In this case, part of the oxygen is simply dissolved in the metal, and part is bound covalently with the metal, that is, it belongs to the formed oxide. The ratio of oxygen bound in the oxide to the total amount of oxygen in the titanium film is related by the factor $k_r(T)$:

$$q_r(x_0,\ t,\ T) = k_r(T) \times q(x_0,\ t) \tag{5}$$

Generally, the factor $k_r(T)$ depends not only on temperature but also on the final product, which is synthesized as a result of the binding of oxygen to titanium. Indeed, we are dealing with polychronous kinetics in this case. However, the high difficulty of separating the pure Magneli phases allows us to conclude that during the formation of oxides with such a close stoichiometry, the reaction rate constants do not differ much from each other. Therefore, we can assume that for all Magneli phases, the factor $k_r(T)$ will be the same.

The band gap energies for various titanium oxides depending on the content of oxygen atoms are shown in Table 2. In the case of the last two values (1.5:1.0 < O:Ti < 2.0:1.0), the band gap $E_g$ can be represented as a linear function of the concentration of bound oxygen $C_r$:

$$E_g = E_g^0\left(4\frac{C_r}{C_r^0} - 3\right) \tag{6}$$

$E_g^0$ and $C_r^0$ are the band gap and the maximum concentration of bound oxygen for TiO$_2$, respectively. Taking Equation (5) into account, we can write the following:

$$\frac{C_r}{C_r^0} = \frac{1}{4}\left(3 + \frac{E_g}{E_g^0}\right) = \frac{q_r/x_0}{q_r^0/x_0} = \frac{k_r(T) \times q}{k_r(T) \times q_0} = \frac{q}{C_0 x_0} \tag{7}$$

Therefore, after simplifying result (4), we obtain the following equation:

$$ln\left(1 - \frac{E_g}{E_g^0}\right) - ln\left(8\frac{x_0}{H}\right) = -\frac{\pi^2 Dt}{H^2} \tag{8}$$

The diffusion factor exponentially depends on the temperature and can be written as the following:

$$D = D_0 \times exp\left(-\frac{E_a}{kT}\right) \tag{9}$$

where $D_0$ is a constant and $E_a$ is the diffusion activation energy. In our case, a simultaneous change in the chemical composition of the oxide occurs; therefore, one needs to assume that the pre-exponential factor $D_0$ will depend on the concentration of bound oxygen. We express the diffusion factor as a product of two functions:

$$D(C_r,\ T) = D_0(C_r) \times exp\left(-\frac{E_a}{kT}\right) \tag{10a}$$

or

$$D(E_g,\ T) = D_0(E_g) \times exp\left(-\frac{E_a}{kT}\right) \tag{10b}$$

since there is a one-to-one correspondence between the band gap energy and the concentration of bound oxygen. At the same time, we must take into account that the solution for the diffusion problem is obtained with the assumption of a constant pre-exponential factor. That is, the function $D_0(E_g)$ must change very little.

Substituting $D(E_g,T)$ from Equation (10b) into Equation (8), we obtain:

$$ln\left(1 - \frac{E_g}{E_g^0}\right) - ln\left(8\frac{x_0}{H}\right) = -\pi^2\frac{D_0(E_g)}{H^2}t \times exp\left(-\frac{E_a}{kT}\right) \tag{11}$$

Then, we substitute the experimental data into Equation (11), taking into account $E_g{}^2 = E_g{}^1$ and $kT_3 = kT_2$, and introduce the notation $L = ln(8x_0/H)$.

$$\begin{cases} ln\left(1 - \frac{E_g^1}{E_g^0}\right) - L = -\pi^2\frac{D_0\left(E_g^1\right)}{H^2}t_1 \times exp\left(-\frac{E_a}{kT_1}\right), & \text{(12a)} \\[2mm] ln\left(1 - \frac{E_g^1}{E_g^0}\right) - L = -\pi^2\frac{D_0\left(E_g^1\right)}{H^2}t_2 \times exp\left(-\frac{E_a}{kT_2}\right), & \text{(12b)} \\[2mm] ln\left(1 - \frac{E_g^3}{E_g^0}\right) - L = -\pi^2\frac{D_0\left(E_g^3\right)}{H^2}t_3 \times exp\left(-\frac{E_a}{kT_2}\right). & \text{(12c)} \end{cases}$$

From Equation (12a,b), we find the effective diffusion activation energy:

$$E_a = ln\left(\frac{t_1}{t_2}\right)\Big/\left(\frac{1}{kT_1} - \frac{1}{kT_2}\right) \tag{13}$$

$$E_a = 0.21\ eV$$

Of course, we should note that this value is related only to the oxide phase with band gap energy of 0.8 eV. Indeed, each Magneli phase has its own diffusion activation energy. But we can expect this calculated value to be similar to ones for other $Ti_nO_{2n-1}$ phases due to the same reasons as discussed in the case of the Equation (5).

Then, we subtract (12a) from (12c) and check how little $D_0(E_g)$ changes:

$$\frac{1}{\pi^2 t_3}\left[ln\left(1 - \frac{E_g^1}{E_g^0}\right) - ln\left(1 - \frac{E_g^3}{E_g^0}\right)\right] \times exp\left(\frac{E_a}{kT_2}\right) = \frac{D_0\left(E_g^3\right)}{H^2} - \frac{D_0\left(E_g^1\right)}{H^2}\frac{t_1}{t_3} \times exp\left(-\frac{E_a}{kT_1} + \frac{E_a}{kT_2}\right) \tag{14}$$

Next, we substitute the experimental data in Equation (14) and obtain the following equation:

$$D_0\left(E_g^3\right) = 0.31 \times D_0\left(E_g^1\right) + 2.96 \times 10^{-5} \times H^2. \tag{15}$$

Since the typical values of $D_0$ in solids are $\sim 10^{-7}$–$10^{-4}$ m$^2$/s and the titanium film thickness is $H \sim 10^{-8}$–$10^{-5}$ m, one can conclude that the term containing $H^2$ will play practically no role. Therefore, we can write the following:

$$D_0\left(E_g^3\right) = 0.31 \times D_0\left(E_g^1\right). \tag{16}$$

Thus, as the oxygen content increases and the band gap broadens, the pre-exponential factor decreases slightly. Such a change is consistent with our assumptions.

## 6. Conclusions

As a result of the above studies, the morphology and local electronic structure of a titanium coating deposited on HOPG surface were established. When interacting with oxygen, $TiO_x$ oxide, where $1.75 < x < 2$, forms on the titanium surface. With an increase in the duration of the annealing procedure for the titanium coating in oxygen, a gradual increase in the band gap of the coating is observed due to a change in the oxygen content in the oxide. A model is proposed describing the change in the band gap of $TiO_x$ ($1.75 < x < 2$) on the surface of a titanium film depending on the duration and temperature of its annealing in oxygen at $P_{O2} \sim 100-200$ Torr. The effective activation energy of oxygen diffusion in titanium oxides $TiO_x$ ($1.75 < x < 2$) is determined to be $E_a = 0.21$ eV. An increase in the oxygen content in the oxide leads to a weak decrease in the pre-exponential factor $D_0$. The proposed theoretical model can be useful for creating titanium oxide coatings with a desired band gap energy by changing the temperature and duration of annealing in oxygen.

**Author Contributions:** Conceptualization, S.Y.S.; methodology, S.Y.S. and A.K.G.; formal analysis, S.A.O.; investigation, A.K.G. and N.V.D.; writing—original draft preparation, V.A.K. and N.V.D.; writing—review and editing, S.Y.S.; visualization, S.A.O. and V.A.K.; supervision, B.R.S.; project administration, M.V.G.; funding acquisition, S.Y.S. All authors have read and agreed to the published version of the manuscript.

**Funding:** The calculation of $TiO_2$ thickness was performed as part of the state assignment of N.N. Semenov Federal Research Center for Chemical Physics, Russian Academy of Sciences (FRCCP RAS), No. 0082-2014-0011 "Nanochemistry", state registration number AAAA-A20-120013190076-0. The rest of the reported study was funded by the Russian Foundation for Basic Research, Grant No.: 18-33-00020.

**Conflicts of Interest:** The authors declare no conflict of interest. The funders had no role in the design of the study; in the collection, analyses, or interpretation of data; in the writing of the manuscript, or in the decision to publish the results.

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
