# Peer review of "Oxidation of Thin Titanium Films: Determination of the Chemical Composition of the Oxide and the Oxygen Diffusion Factor"

_crystals, doi:10.3390/cryst10020117_

Round 1
Reviewer 1 Report
The paper from Sarvadiy et al. deals with the investigation of titanium oxides formed on titanium coatings deposited on highly oriented pyrolytic graphite. The authors findings are that the duration and temperature of its annealing in oxygen lead to a gradual change in the band gap of the formed TiOx (1.75 < x < 2) on the surface of a titanium film. Albeit interesting, the experimental data do not seem convincing for the claims aiming at the possibility to tune the band gap value. On the contrary, the authors produce a rigorous theoretical model to better describe the role of oxygen diffusion in the titanium layer leading to the formation of oxides species.
The authors should carry out several changes to the manuscript before it could be considered:
In the introduction, they have to cite the papers J. Phys. Chem. B 2005, 109, 51, 24411-24426 and Solar Energy, Volume 83, Issue 9, September 2009, Pages 1499-1508. Especially the first one explored the various chemical compositions of the films prepared by reactive evaporation of Ti in oxygen. The results of this paper are quite exhaustive in describing the chemistry of the TiOx phases. The STM image in Figure 2 is of low quality. Please provide another one of higher quality. Rephrase the claims of the article, since there is not a clear demonstration of the band gap tuning but rather an interesting theoretical modelling for understanding the phenomenon.
Only after such amendments, the paper could be considered for the journal.
Author Response
Many thanks for your valuable comments that helped us to improve the quality of the manuscript.
1) In the introduction, they have to cite the papers J. Phys. Chem. B 2005, 109, 51, 24411-24426 and Solar Energy, Volume 83, Issue 9, September 2009, Pages 1499-1508. Especially the first one explored the various chemical compositions of the films prepared by reactive evaporation of Ti in oxygen. The results of this paper are quite exhaustive in describing the chemistry of the TiOx phases.
Answer: Articles mentioned by you are significant indeed. Both of them are excellent examples of multifaceted approach to the investigation of the oxidation of ultrathin metal films. Many thanks for these valuable references. We have included them in the manuscript (Introduction).
2) The STM image in Figure 2 is of low quality. Please provide another one of higher quality.
Answer: Figure 2 has been substituted by another one with higher resolution.
3) Rephrase the claims of the article, since there is not a clear demonstration of the band gap tuning but rather an interesting theoretical modelling for understanding the phenomenon.
Answer: We agree with this amendment. Relevant changes have been added to the manuscript (last sentence in Abstract and last sentence in Conclusion).
Reviewer 2 Report
The work is quite interesting. An original technology for producing of films of the TiO2 type is described, which makes it possible to obtain structures with a given band gap Eg. A theoretical model of film formation during oxygen diffusion through a TiO2 film is presented.
Remarks:
It is not clear what material is obtained with this technology. For example, it is possible to obtain material in the form of a mixture of TiO2 anatase fractions and Ti metal particles in the ratio x / 1-x, or as mixed crystals (TiO2) xTi (1-x). The band gap, measured by the indirect method, is correlated with the literature on the band gap of the “pure” TiO2 anatase, which are also determined by the indirect method. There are experimental data [1] obtained by “direct” methods from the UV absorption and photoconductivity spectra that for a thin film of “pure” TiO2 anatase, the band gap is 3.7 - 3.8 eV. The discrepancy in the Eg TiO2 value affects the interpretation of the results, for example, the maximum Eg of the resulting films and, accordingly, xmax.
V. I. Bredikhin, V. N. Burenina, Yu. A. Mamayev, S. N. Yashin, S. (2013). Spectral and Relaxation Properties of the Photoconductivity of Thin TiO2 Films Produced by the Sol-gel Technique. Physical Science International Journal, 3(4), 642-665. http://www.journalpsij.com/index.php/PSIJ/article/view/23149, DOI : 10.9734/prri/2013/4253.
Author Response
Many thanks for your valuable comments that helped us to improve the quality of the manuscript.
1) It is not clear what material is obtained with this technology. For example, it is possible to obtain material in the form of a mixture of TiO2 anatase fractions and Ti metal particles in the ratio x / 1-x, or as mixed crystals (TiO2) xTi (1-x).
Answer: Since STM is sensitive to the surface of the sample, we can’t say anything about inherent crystal structure. Calculation presented in Determination of the oxide chemical composition let us conclude that synthesized coating shouldn’t be considered as mixture of TiO2 fractions and Ti metal particles. Relevant clarification has been added to the manuscript (p.7). We suppose that a range of similar oxide phases is formed on the sample surface due to titanium oxidation. You are likely to be right saying about mixed crystals, but we need to take into account Magneli phases rather than pure metal (p.8). That is, such crystals composition seems to be complicated like Ti3O5(x)×Ti4O7(y)×…×TiO2(1–x–y…).
2) The band gap, measured by the indirect method, is correlated with the literature on the band gap of the “pure” TiO2 anatase, which are also determined by the indirect method. There are experimental data [10.9734/prri/2013/4253] obtained by “direct” methods from the UV absorption and photoconductivity spectra that for a thin film of “pure” TiO2 anatase, the band gap is 3.7 - 3.8 eV. The discrepancy in the Eg TiO2 value affects the interpretation of the results, for example, the maximum Eg of the resulting films and, accordingly, xmax.
Answer: You are absolutely right saying that such discrepancy should be taken into account. Relevant clarification has been added to the manuscript (p.7-8). STS results show us that there are points on the sample surface where local Eg exceeds even 3.7-3.8 eV. Such high values can appear due to the charging effects in the volume of the formed dioxide. But we don’t see any clear peak neither at 3.0-3.2 eV nor at 3.7-3.8 eV. So, indeed we cannot distinguish pure TiO2 band gap and TiO2 band gap with charging effects without literary data and make any strict attributions due to its little quantity. But maxima of our histograms fall at 0.8 and 1.8 eV – they are far from both variants of dioxide band gap energy. That is, in the case of our experiments the interpretation of maxima position doesn’t depend on the true value of titanium dioxide Eg. At the same time we decided to make no changes in calculations, since the dioxide true value isn’t included in final formula (13). And in formulae (14-15b) it is included only in the negligible term. That is, this amendment plays no role in the case of calculation.
Round 2
Reviewer 1 Report
The authors have significantly improved the quality of the manuscript, which is now ready for publication.